# Enhanced Photodynamic Efficacy Using 1,8-Naphthalimides: Potential Application in Antibacterial Photodynamic Therapy

**DOI:** 10.3390/molecules27185743

**Published:** 2022-09-06

**Authors:** Desislava Staneva, Awad I. Said, Evgenia Vasileva-Tonkova, Ivo Grabchev

**Affiliations:** 1Department of Textile, Leader and Fuels, University of Chemical Technology and Metallurgy, 1756 Sofia, Bulgaria; 2Chemistry Department, Faculty of Science, Assiut University, Assiut 71516, Egypt; 3Department of Chemistry and Biochemistry, Physiology and Pathophysiology, Faculty of Medicine, Sofia University “St. Kliment Ohridski”, 1407 Sofia, Bulgaria; 4The Stephan Angeloff Institute of Microbiology, Bulgarian Academy of Sciences, 1113 Sofia, Bulgaria

**Keywords:** 1,8-naphthalimides, photodynamic therapy, antibacterial, textile

## Abstract

This study addresses the need for antibacterial medication that can overcome the current problems of antibiotics. It does so by suggesting two 1,8-naphthalimides (NI1 and NI2) containing a pyridinium nucleus become attached to the imide-nitrogen atom via a methylene spacer. Those fluorescent derivatives are covalently bonded to the surface of a chloroacetyl-chloride-modified cotton fabric. The iodometric method was used to study the generation of singlet oxygen (^1^O_2_) by irradiation of KI in the presence of monomeric 1,8-naphthalimides and the dyed textile material. Both compounds generated reactive singlet oxygen, and their activity was preserved even after they were deposited onto the cotton fabric. The antibacterial activity of NI1 and NI2 in solution and after their covalent bonding to the cotton fabric was investigated. In vitro tests were performed against the model gram-positive bacteria *B. cereus* and gram-negative *P. aeruginosa* bacteria in dark and under light iradiation. Compound NI2 showed higher antibacterial activity than compound NI1. The light irradiation enhanced the antimicrobial activity of the compounds, with a better effect achieved against *B. cereus*.

## 1. Introduction

In recent years, bacteria’s resistance to the antibiotics used in clinical practice has increased to dangerously high levels in many parts of the world [1,2]. New mechanisms of resistance have emerged and spread, leading to an inability to treat common infectious diseases because the antibiotics administered are becoming less effective. This problem necessitates a search for new compounds with good antimicrobial activity, as well as the invention of new strategies for their usage. In this regard, cyclic imides are of particular interest because of their biologically active compounds with well-defined antitumor and microbiological activity. Special attention has been paid to the 1,8-naphthalimide derivatives, which are a type of cyclic amide with well-defined hydrophobicity and a п-conjugated aromatic skeleton [3,4,5,6]. These compounds have a compact structure and, depending on the type of substituents, can drastically change their photophysical properties. The most common substituents in the structure of 1,8-naphthalimides are various substituted alkoxy and amino groups, which give them blue or yellow-green fluorescence. Compounds with such properties are at the heart of the design of optical sensors for the detection of metal cations, anions, neutral molecules, and enzymes [7,8,9]. They are also used for dyeing textile and polymeric materials [10], and in organic light-emitting diodes [11], etc.

Antimicrobial photodynamic therapy is a new method for inactivating a wide range of microorganisms that are highly resistant to antimicrobial substances that are used in practice [12,13,14]. The method uses special compounds called photosensitizers (PS), which, during irradiation with sunlight in the presence of molecular oxygen (O_2_), generate highly cytotoxic reactive oxygen species (ROS) and, in particular, singlet oxygen (^1^O_2_) that reacts with the cytoplasmic membrane and the cell walls of microorganisms. This inactivates them, eliminating the possibility of photoresist strains occurring. Phenothiazine, xanthene dyes, porphyrins, and phthalocyanines have been used as PS [15]. Fullerene and metal nanoparticles, such as Ag, Au, Pt, etc., have been implemented as very good antibacterial agents. Therefore, there is increased interest in the search for new, effective PS. To the best of our knowledge, there are no data on the investigation of 1,8-naphthalimides as PS for antibacterial photodynamic inactivation. In our laboratory, we have begun systematic studies utilizing 1,8-naphthalimides as PS in the inactivation of Gram-positive and Gram-negative bacteria, both when employing only the compounds, and in cases after their deposition onto cotton fabric [16,17,18,19]. In these cases, 1,8-naphthalimides have been attached to the periphery of low-generation dendrimers. The photoactive compounds are attached to the cotton surface mainly by hydrogen bonds and Van der Waals interactions.

This paper describes the synthesis and photophysical characterization of two 1,8-naphthalimides and their covalent bonding to the cotton fabric. The antimicrobial activity of the compounds has been investigated against Gram-positive and Gram-negative bacteria in solution and applied on cotton fabric in the dark and after light irradiation. 

## 2. Results and Discussion

### 2.1. Synthesis of 1,8-Naphthalimide Derivatives

In order to obtain antibacterial textiles, two 1,8-naphthalimide derivatives emitting blue and yellow-green fluorescence were synthesized (Figure 1). Compound NI1 was prepared according to the modified procedures [20,21] by reacting 1,8-naphthalic anhydride (NA) with 2-aminomethylpyridine in ethanol solution at 60 °C. Upon cooling, a precipitate of pale-yellow crystals formed, which was isolated from the reaction medium by filtration. Compound NI2 was obtained via a two-step synthesis during which a 4-nitro-1,8-naphthalic anhydride (NNA) was reacted with a 2-aminomethylpyridine following a method analogous to the one for NI1 preparation. The final NI2 product was obtained after nucleophilic substitution of the nitro group with an *N*,*N*-dimethylamino group in a DMF solution at room temperature, run for 24 h. The chemical structure of the synthesized compounds was confirmed by IR, ^1^H-NMR and ^13^C-NMR spectra (Appendix A) and elemental analysis. 

### 2.2. Photophysical Characteristics

The photophysical characteristics of 1,8-naphthalimide derivatives depend strongly on the polarization of their chromophore structure [22]. The distribution of the electron density in each chemical structure is unique, which determines its specific fluorescent properties and characteristics. The main photophysical characteristics of fluorophores that are determined and calculated from experimental data are the absorption (λ_A_) and fluorescent (λ_F_) maximum, molar absorbency (ε), Stokes shift (υ_A_ − υ_F_), and quantum yield of fluorescence (Φ_F_). In order to study the synthesized 1,8-naphthalimides as photosensitizers of antibacterial photodynamic therapy, their functional characteristics in organic solvents with different polarities were studied in detail. The data obtained for NI1 and NI2 are collected in Table 1 and Table 2. Due to the absence of an electron donor substituent in NI1, the polarization of the molecule was weak. It absorbs in the UV region at 333–350 nm and emitted blue fluorescence with maxima at 378–398 nm. For NI2 having an electron-donating substituent N(CH_3_)_2_ the both maxima were batochromically shifted and they are in the visible spectral region at 412–435 nm and 500–540 nm respectively.

Figure 1 shows the dependence of the position of the absorption and fluorescent maximum of NI2 on the polarity of the solvents and the observed positive solvatochromism. The nature of the solvents significantly affected the spectral properties of the compound, on the one hand through their polarity, and on the other hand, through the possibility of specific dipole–dipole interactions leading to a change in the polarity of the chromophore system. A similar dependence was observed in the case of NI1.

The quantum efficiencies of NI1 and NI2 were found by calculating their quantum fluorescence yields on the basis of their absorption and fluorescence spectra using standards as quinine bisulfate/H_2_SO_4_ (Φ_F_ = 0.546) for NI1 and Rhodamine 6G (Φ_F_ = 0.96) for NI2. The results are summarized in Table 1 and Table 2. The low values of the quantum yield at NI1 were due to the weak polarization of the chromophore system due to the absence of an electron donor substituent at the C-4 position of the 1,8-naphthalimide structure. In the case of NI2, the weak fluorescence emission can be explained by a change in the coplanarity of the molecule resulting from spatial factors, such as rotation around the N-CH3 bonds at position C-4, or from the interaction of one from methyl groups with the hydrogen atom at the C-5 position of the naphthalene nucleus [23,24,25].

### 2.3. Molecular Logic Behavior of Probes NI1 and NI2 by H^+^ and HO^−^ as Inputs

As seen in Figure 2A, probe NI2 alone (C = 3 × 10^−5^ mol L^−1^, ethanol/water (1:1)) gave high fluorescence emission at 535 nm (λ_ex._ = 430 nm) (coded for binary 1). When adding input H^+^ alone (10^−2^ M, decreasing pH to 2), the emission decreased, and the probe turned into a low state (coded for binary 0). However, by adding input HO^−^ alone (10^−2^ M, increasing pH to 12), the probe kept its fluorescence emission high (coded for binary 1). Finally, the simultaneous presence of the two inputs (H^+^ and HO^−^) annihilated each other, keeping the emission high (coded as 1). The behavior of the probe in terms of the fluorescence emission at 535 nm as output and H^+^ and HO− as inputs mimicked INHIBIT logic gate [26] (Table 3).

On the other hand, as shown in Figure 3B, probe NI1 (C = 3 × 10^−5^ mol L^−1^, ethanol/water (1:1)) exhibited low emission at 390 nm (λ_ex._ = 340 nm) in the presence of either H^+^ or HO^−^ as an input. However, the absence or presence of both H^+^ and HO^−^ kept the fluorescence emission high. This behavior mimicked the XNOR logic gate [27], Table 3.

### 2.4. Dyeing of Modified Cotton Fabric with NI1 and NI2

Cotton fabric is a material that is commonly used to make various antibacterial products that are used in clinical practice. In order to prevent the formation of bacterial biofilms, various methods are used for the surface modification of such materials [28]. NI1 and NI2 contain a pyridinium nucleus, which is attached to the imide-nitrogen atom via a methylene (-CH_2_-) spacer, which allows it to be quaternized. For this purpose, cotton fabric was modified with chloroacetyl chloride. Thus, the reactivity of the chlorine atom was enhanced by C=O from the ester group after it was introduced onto the material surface [29]. The dyeing of the modified cotton fabric with NI1 or NI2 was carried out at 80 °C in a DMF medium for 3 h, wherein the 1,8-naphthalimides were covalently linked by quaternization of the pyridine nucleus (Figure 2). The dyed fabrics were removed, rinsed with water, and then treated with boiling ethanol for 60 min to remove the unreacted 1,8-naphthalimides from the cotton surface. In this way, binding the fluorophores to the cotton fabric through a strong covalent bond provides a lasting effect and prevents their migration from the surface [10].

### 2.5. Colorimetric Characteristics of Cotton Fabrics Treated with Dendrimers P1 and P2

The color characteristics of the cotton fabrics treated with NI1 and NI2 were determined by CIEab parameters (L*, a* and b*) and the chromaticity coordinates (x and y), which were compared to the initial untreated cotton. The data are summarized in Table 4. The results showed that the cotton fabrics treated with NI2 had a brilliant yellow color, while the use of NI1 did not lead to a change in the cotton fabric color. These data were also confirmed by the color differences ΔE* = 2102 for NI1 and ΔE* = 30.748 for NI2, respectively, for the untreated cotton fabric used as a standard ΔE* = 0.

### 2.6. Photo-Oxidation Studies of NI1 and NI2 and on Dyed Cotton Fabric

The generation of singlet oxygen from 1,8-naphthalimide derivatives and dyed cotton fabrics in an aqueous solution was proven by the iodometric method [30,31,32]. Figure 3A shows the absorption spectra of irradiated KI solution in the presence of NI2 taken in the spectral range of 270–500 nm. The non-irradiated solution showed no absorption in this spectral region. The spectra recorded after irradiation show the appearance of two absorption maxima in the UV region, one at 288 nm and the other at 352 nm. The two absorption maxima are typical of I_3_^−^, which form during the photo-oxidation of I^−^ via KI from formed reactive singlet oxygen (^1^O_2_) [30,31,32]. The same spectral behavior was observed when using a solution of NI1. The absorption intensity at these maxima increased with the time of irradiation, which demonstrated, respectively, an increase in the amount of singlet oxygen generated.

The dependence of the absorbance at 352 nm as a function of the irradiation time of NI1 and NI2 is shown in Figure 3B, which reveals that the activity of NI2 was higher than that of NI1. The better activity of NI2 is probably due to the fact that the maximum of its absorption is in the visible spectral range. The absorption of the solutions in the dark, as well as the irradiated solutions of NI in the absence of KI, were also studied. No absorption of these solutions was registered, indicating that no singlet oxygen was generated in the dark.

The cotton fabrics dyed with NI1 and NI2 were also tested to generate singlet oxygen under the same conditions as the monomeric 1,8-naphthalimides. Figure 4A shows the dependence of the absorption of the aqueous solution of KI with the irradiation time in the presence of fabric dyed with NI2. The results showed the same spectral dependence as for NI2, which means that 1,8-naphthalimides which covalently bound to the cotton surface retain their ability to generate singlet oxygen. The activity of the two colored fabrics was almost the same, as shown in Figure 4B.

In the case of virgin cotton fabric and cotton fabric dyed with NI1 and NI2, studied in the dark, no absorption was registered.

### 2.7. Effect of Light Irradiation on Bacterial Growth

The effect of light on the antimicrobial activity of the investigated compounds alone and applied to cotton fabric was tested in a liquid medium against model bacteria *B. cereus* and *P. aeruginosa*. The results showed that NI2 inhibited the growth of the tested cultures more effectively than NI1, compared to the negative control. Gram-positive *B. cereus* was found to be more sensitive than Gram-negative *P. aeruginosa*. The antimicrobial effect was better expressed under light irradiation than in the dark, and higher growth inhibition of the strains was produced by NI2 than by NI1 (Figure 5). Against *B. cereus* (Figure 5a), at a concentration of 250 µg/mL of NI1, about 63% growth reduction was established in the dark, and almost complete growth inhibition was established in the illuminated sample. In the presence of NI2 at a concentration of 20 µg/mL, growth inhibition in the dark was about 75%, whereas it was almost complete under light irradiation. Gram-negative *P. aeruginosa* showed higher resistance to the compounds than *B. cereus* (Figure 5b). At a concentration of 350 µg/mL, NI1 inhibited the growth of *P. aeruginosa* by about 38% in the dark and about 52% under light. At the same concentration, growth inhibition by NI2 in dark and under light was about 63% and 86%, respectively.

The increased antimicrobial activity of the studied photoactive dendrimers under light irradiation can be explained by their ability to bind to a bacterial membrane and generate highly reactive singlet oxygen (^1^O_2_) particles upon photostimulation [33]. These reactive particles attack the external layer of the bacterial membrane by multi-target action. Thus, oxidative stress causes irreparable damage to the cellular bacterial components, leading to their inactivation [34].

The results from the experiments with non-treated cotton fabric and cotton fabrics treated with compounds NI1 and NI2 are presented in Figure 6. As can be seen, under light irradiation both treated cotton fabrics significantly inhibited the growth of *B. cereus* (more than 80%) in comparison with this effect in the dark (Figure 6a). A slight growth reduction under light was established in *P. aeruginosa* (Figure 6b): about 6% for NI1 and about 15% for NI2. The compounds are attached to the cotton surface by covalent bonds. Direct contact of bacterial cells with the cotton surface mainly contributed to the antimicrobial effect of the treated cotton fabrics.

## 3. Experimental Section

### 3.1. Synthesis of 2-(Pyridin-4-ylmethyl)-1H-benzo[de]isoquinoline-1,3(2H)-dione (NI1)

2-aminomethylpyridine 1.0 mL. (0.01 M) was added dropwise over 60 min at 60 °C to a solution of 1.97 g (0.01 M) 1,8-naphthalic anhydride in ethanol, then stirred for another 60 min under these conditions. After cooling, the final product was isolated as a precipitate of needle crystals. Yield 98%, 2.79 g.

FTIR cm^−1^:3054, 2962, 1700,1655, 1630, 1586, 1487, 1433, 1379, 1334, 1234, 1174, 1070, 950, 853, 775, 647. 

^1^H-NMR (DMSO-d_6_, 600 MHz, δ (ppm): 8.52 (dd *J* = 7.35 Hz, 2H, Ar-H), 844 (dd *J* = 4.23 Hz, 1H, Ar-H), 8.14 (dd *J* = 8.42 Hz, 1H, Ar-H), 7.67 (t, *J* = 7.86 Hz, 1H, Ar-H), 7.28 (dd, *J* = 3.96 Hz, 1H, Ar-H), 7.17 (bs, 2H, Ar-H), 5.27 (s, 2H, CH_2_).

^13^C NMR (DMSO-d_6_, 150 MHz, δ (ppm): 164.3, 150.2, 146.2, 135.6, 131.9, 128.7, 127.2, 122.4, 42.8.

Analysis: C_18_H_9_N_2_O_2_ (285.2) Calc: C-75.73; H-3.16; N-9.81 Found: C-75.61; H-3.28; N-9.96.

### 3.2. Synthesis of 6-Nitro-2-(pyridin-4-ylmethyl)-1H-benzo[de]isoquinoline-1,3(2H)-dione (NI0)

2-aminomethylpyridine 1.00 mL. (0.01 M) was added dropwise over 60 min at 60 °C to a solution of 2.43 g (0.01 M) 4-nitronaphthalic anhydride in ethanol, then stirred for another 60 min under these conditions. After cooling, the liquid was added to 200 mL of water and the precipitate was filtered out. Yield 79%, 2.61 g.

^1^H NMR (500 MHz, DMSO) δ 8.72 (d, *J* = 8.6 Hz, 1H), 8.63 (dd, *J* = 11.4, 7.6 Hz, 2H), 8.56 (d, *J* = 8.0 Hz, 1H), 8.48 (d, *J* = 5.6 Hz, 2H), 8.10 (t, *J* = 8.0 Hz, 1H), 7.38 (d, *J* = 5.8 Hz, 2H), 5.27 (s, 2H). ^13^C NMR (126 MHz, DMSO) δ 163.6, 162.8, 150.1, 149.8, 146.3, 132.4, 130.4, 130.4, 129.5, 129.1, 127.0, 124.7, 123.3, 123.1, 122.6, 43.0.

Analysis: C_18_H_8_N_3_O_4_ (330.2) Calc: C-65.41; H-2.43; N-12.72 Found: C-65.50; H-2.48; N-12.80.

### 3.3. Synthesis of 6-(Dimethylamino)-2-(pyridin-4-ylmethyl)-1H-benzo[de]isoquinoline-1,3(2H)-dione (NI2)

The compound NI 0 (1.65 g, 0.005 M) was dissolved in 20 mL of DMF, and 0.101 mL (0.005 M) of dimethylamine was added. After that, the mixture was stirred for 24 h at 25 °C. The liquid was poured into 200 mL of water and the formed precipitate was filtered. Yield 81%, 1.33 g.

FTIR cm^−1^: 3031, 2953, 1686, 1650, 1579, 1520, 1379, 1333, 1239, 1132, 1129, 784, 759

^1^H NMR (500 MHz, DMSO) δ 8.72 (d, *J* = 8.6 Hz, 1H), 8.63 (dd, *J* = 12.2, 7.6 Hz, 2H), 8.55 (dd, *J* = 12.6, 8.3 Hz, 2H), 8.46 (d, *J* = 7.2 Hz, 1H), 8.11 (t, *J* = 8.0 Hz, 1H), 7.39 (d, *J* = 4.4 Hz, 2H), 5.24 (s, 2H), 3.12 (s, 6H). ^13^C NMR (126 MHz, DMSO) δ 163.6, 162.8, 157.5, 150.1, 146.3, 133.6, 132.4, 131.4, 130.4, 129.6, 129.2, 127.0, 125.5, 124.7, 123.2, 122.5, 113.4, 44.9, 43.0.

Analysis: C_20_H_14_N_3_O_2_ (328.3) Calc: C-65.41; H-2.43; N-12.72 Found: C-65.50; H-2.48; N-12.80.

### 3.4. Dyeing the Cotton Fabric with NI1 and NI2

A sample of 1.0 g of cotton fabric (140 g/m^2^) was dipped into 40 mL DMF. NI1 or NI2 (0.25 mg dissolved in 10 mL DMF) was added slowly and then stirred at 80 °C. After dyeing for 3 h, the sample was washed with deionized water and dried. The dyed fabrics were removed and treated with boiling ethanol for 60 min. Fluorescent spectroscopy studies revealed that 0.22 mg of (NI1) and 0.23 mg of (NI2) were covalently bonded to the cotton surface.

### 3.5. Materials and Methods

Thermo Spectronic Unicam UV 500 spectrophotometer was used for UV-Vis spectrophotometric investigations. Emission spectra were taken on a “Cary Eclipse” spectrofluorometer. The color coordinates (L*a*, b*, XYZ and xy) dyed with NI1 and NI2 were determined on a Datacolor Spectraflash SF300 spectrophotometer (Datacolor, NJ, USA). All organic solvents used in this study were of spectroscopic grade ^1^H NMR (500.3 MHz) and ^13^C (166.7 MHz) spectra were acquired from a BRUKER AVANCE III spectrometer.

### 3.6. In Vitro Antimicrobial Assay

The antimicrobial activity of the investigated compounds was tested in meat-peptone broth (MPB) against Gram-positive *Bacillus cereus* and Gram-negative *Pseudomonas aeruginosa* which were used as model strains. The tests were performed in the presence of light and in the dark. The compounds were dissolved in DMSO at an initial concentration of 5.0 mg/mL and further diluted in test tubes with MPB to final concentrations in the range of 10–400 µg/mL. The inocula were prepared by diluting the overnight cultures with 0.9% NaCl to a 0.5 McFarland standard. The tubes were inoculated with each standardized cell suspension and incubated at an appropriate temperature for 18 h in the presence of light and in dark. Positive controls (compounds and MPB, without inoculum) and negative controls (MPB and inoculum, without compounds) were also prepared. Microbial growth was determined by the turbidity of the medium at 600 nm (OD_600_). All assays were performed in triplicate and the average was taken (standard deviations are less than 5%).

The antimicrobial activity of cotton fabrics treated with the compounds NI1 and NI2 was tested in MPB containing Gram-positive *Bacillus cereus* or Gram-negative *Pseudomonas aeruginosa* under light irradiation and in the dark. Tubes containing MPB and square-shaped cotton specimens (10 mm × 10 mm) were inoculated with standardized microbial suspension. Tubes with native cotton and without specimens were also prepared as controls. Two sets of tubes were prepared for the tests in the presence and absence of light. After 18 h incubation at an appropriate temperature, the specimens were removed and OD_600_ was determined. The antimicrobial activity of the samples was evaluated by the reduction in bacterial growth in the presence of the treated specimens compared to native specimens. All assays were performed in triplicate and the averages are given (standard deviations less than 5%).

A Xe lamp 150 W with a spectral window of the solar simulator (185–1100 nm) was used for irradiation of all antibacterial studies at a distance of 25 cm from the samples.

### 3.7. Iodometric Absorption Measurements 

An aqueous solution of KI (10 mL, 0.5 M), and NI1 and NI2 at a concentration of 1 × 10^−6^ M, or cotton fabrics dyed with them (1 cm^2^) were illuminated for 60 min (Xe lamp 150 W at a 25 cm sample distance, a spectral window of the solar simulator: 185–1100 nm). The absorption spectra of the solution were recorded at intervals of 5 min.

## 4. Conclusions

The synthesis and photophysical characterization of two reactive 1,8-naphthalimide derivatives emitting yellow-green (NI1) and blue (NI2) fluorescence were described. The photophysical properties were determined in organic solvents of different polarities. The new compounds were used to dye cotton fabrics, which resulted in covalent bonding of 1,8-naphthalimides to the cotton surface, whereby the resulting fabrics were white and yellow. The monomeric 1,8-naphthalimides and the cotton fabrics treated with them were found to generate reactive singled oxygen during irradiation with KI solution in sunlight. The antibacterial activity of both compounds and the dyed cotton fabrics was tested in vitro against Gram-positive *B. cereus* and Gram-negative *P. aeruginosa* in the dark and after light irradiation. Compound NI2 was found to be more active against the studied pathogens compared to NI1. The antimicrobial activity of the new agents is enhanced by light irradiation. The compounds exhibited better-expressed antimicrobial action against *B. cereus*. These results suggest the potential of the novel compounds for usage are effective antibacterial agents.

## Data Availability

Not applicable.

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
