# Peer review of "Enhanced Photodynamic Efficacy Using 1,8-Naphthalimides: Potential Application in Antibacterial Photodynamic Therapy"

_molecules, 2022, doi:10.3390/molecules27185743_

Round 1
Reviewer 1 Report
This approach to photo-inactivation involves use of a fabric that contains either of two photosensitizing agents. These agents have absorbance optima in the vicinity of 345 and 420 nm. Some of the data are interesting but not relevant to photoinactivation, e.g., effects of dielectric constant on parameters. Fluorescence emission spectra are shown but also irrelevant in the context of photoinactivation.
Cotton fabrics are treated with these agents. This is reported to lead to the formation of a yellow color with one of the agents. Irradiation of fabrics containing these agents results in formation of reactive oxygen species. It is not clear how this was done. One can infer that the fabrics were placed in a KI solution and the absorption spectra measured as a function of time during irradiation, but this is not specified.
Methods used for assessing anti-microbial action of these agents are not indicated. Line 191 discusses liquid media but this is the only indication. It is also not obvious what OD600 means. Assuming that this reflects the viability of the microorganisms, there appears to be some dark toxicity of the preparations. Is there an explanation for this? Fig. 5 reports that irradiation is more effective for treatment of B. cereus than for P. aeruginosa. So it appears that some microorganisms will be less affected than others; perhaps not a surprise. What does ‘visible light’ mean? There needs to be a better description of the wavelength and light dose in the figure legends.
The text of page 8 indicates that Fig. 6 deals with non-treated and treated fabrics. But the legend to the figure does not indicate which fabrics were treated and which not treated with the photosensitizers.
Other points: Why the treatment boiling ethanol (line 266)? The ‘experimental part’ contains no details concerning the carrying out of experiments, How were the microorganisms applied to the fabrics? Under what conditions were the fabrics irradiated: wet or dry? How was microbial survival determined? There are no details concerning exactly how this procedure would be used. I doubt that people are going to be dying their textiles to prevent microbial infections.
Author Response
This approach to photo-inactivation involves use of a fabric that contains either of two photosensitizing agents. These agents have absorbance optima in the vicinity of 345 and 420 nm. Some of the data are interesting but not relevant to photoinactivation, e.g., effects of dielectric constant on parameters. Fluorescence emission spectra are shown but also irrelevant in the context of photoinactivation.
Answer:
The use of the photophysical characteristics of the synthesized 1,8-naphthalimides aims to give a complete characterization of these photoactive compounds
Cotton fabrics are treated with these agents. This is reported to lead to the formation of a yellow color with one of the agents. Irradiation of fabrics containing these agents results in formation of reactive oxygen species. It is not clear how this was done. One can infer that the fabrics were placed in a KI solution and the absorption spectra measured as a function of time during irradiation, but this is not specified.
Answer:
The explanation has been done in the text.
Methods used for assessing anti-microbial action of these agents are not indicated. Line 191 discusses liquid media but this is the only indication. It is also not obvious what OD600 means. Assuming that this reflects the viability of the microorganisms, there appears to be some dark toxicity of the preparations. Is there an explanation for this? Fig. 5 reports that irradiation is more effective for treatment of B. cereus than for P. aeruginosa. So it appears that some microorganisms will be less affected than others; perhaps not a surprise. What does ‘visible light’ mean? There needs to be a better description of the wavelength and light dose in the figure legends.
Answer:
Methods used for assessing antimicrobial action of the compounds are included in the text and explanations are given.
The text of page 8 indicates that Fig. 6 deals with non-treated and treated fabrics. But the legend to the figure does not indicate which fabrics were treated and which not treated with the photosensitizers.
Answer:
Was added in the legend
Other points: Why the treatment boiling ethanol (line 266)? The ‘experimental part’ contains no details concerning the carrying out of experiments, How were the microorganisms applied to the fabrics? Under what conditions were the fabrics irradiated: wet or dry? How was microbial survival determined? There are no details concerning exactly how this procedure would be used. I doubt that people are going to be dying their textiles to prevent microbial infections.
Answer:
Ethanol is a good solvent in the synthesis of the condensation of 1,8-naphthoic anhydride with primary amines and for this reason, we use it.
The microbiological explanation was added in the text
Reviewer 2 Report
Desislava Staneva et al. has investigated enhanced photodynamic efficacy using 1,8-naphthalimides: potential application in antibacterial photodynamic therapy.
The study addresses the need for antibacterial medication able to overcome the current problem of antibiotics by suggesting two novel 1,8-naphthalimides (NI1 and NI2), containing a pyridinium nucleus attached to the imide nitrogen atom via a methylene spacer.
In general the manuscript contain relevant paragraphs that have been discussed, and the research has been conducted in a proper way. The selection of bibliography is appropriate to the content of the manuscript. In the conclusion, the authors have included short thoughts from the research presented. Figures and tables are well presented with adequate quality and resolution.
After close evaluation of the paper I suggest revision according to the next points:
1. Please provide FTIR and 1H-NMR and 13C-NMR spectra for the following part of the study:
“3. Experimental part
3.1. Synthesis of 2-(pyridin-4-ylmethyl)-1H-benzo[de]isoquinoline-1,3(2H)-dione (NI1 )
2-aminomethylpyridine 1.0 ml. (0.01M) was added dropwise over 60 min. at 60°C to a solution of 1.97g (0.01M) 1,8-naphthalic anhydride in ethanol, then stirred for another 60 min under these conditions. After cooling, the final product was isolated as a precipitate of needle crystals. Yield 98%, 2.79g.
3.2. Synthesis of 6-nitro-2-(pyridin-4-ylmethyl)-1H-benzo[de]isoquinoline-1,3(2H)-dione (NI0)
2-aminomethylpyridine 1.00 ml. (0.01M) was added dropwise over 60 min at 60°C to a solution of 2.43 g (0.01M) 4-nitro-naphthalic anhydride in ethanol, then stirred for another 60 min under these conditions. After cooling, the liquid has been added to 200ml water and the precipitate has been filtered out. Yield 79%, 2.61g.
3.3. Synthesis of 6-(dimethylamino)-2-(pyridin-4-ylmethyl)-1H-benzo[de]isoquinoline-1,3(2H)-dione (NI2)
The compound NI 0 (1.65 g ,0.005 M) was dissolved in 20 ml of DMF, and 0.101 ml (0.005 M) of dimethylamine was added and after that and the mixture is stirred for 24 250 hours at 25 ° C. The liquid was poured into 200ml water and filtered the formed precipitate. Yield 81%, 1.33g.”
2. Part of the discussion should be completed.
3. Whether the authors were tempted to do statistical analysis?
Author Response
Desislava Staneva et al. has investigated enhanced photodynamic efficacy using 1,8-naphthalimides: potential application in antibacterial photodynamic therapy.
The study addresses the need for antibacterial medication able to overcome the current problem of antibiotics by suggesting two novel 1,8-naphthalimides (NI1 and NI2), containing a pyridinium nucleus attached to the imide nitrogen atom via a methylene spacer.
In general the manuscript contain relevant paragraphs that have been discussed, and the research has been conducted in a proper way. The selection of bibliography is appropriate to the content of the manuscript. In the conclusion, the authors have included short thoughts from the research presented. Figures and tables are well presented with adequate quality and resolution.
After close evaluation of the paper I suggest revision according to the next points:
1. Please provide FTIR and 1H-NMR and 13C-NMR spectra for the following part of the study:
“3. Experimental part
3.1. Synthesis of 2-(pyridin-4-ylmethyl)-1H-benzo[de]isoquinoline-1,3(2H)-dione (NI1 )
2-aminomethylpyridine 1.0 ml. (0.01M) was added dropwise over 60 min. at 60°C to a solution of 1.97g (0.01M) 1,8-naphthalic anhydride in ethanol, then stirred for another 60 min under these conditions. After cooling, the final product was isolated as a precipitate of needle crystals. Yield 98%, 2.79g.
3.2. Synthesis of 6-nitro-2-(pyridin-4-ylmethyl)-1H-benzo[de]isoquinoline-1,3(2H)-dione (NI0)
2-aminomethylpyridine 1.00 ml. (0.01M) was added dropwise over 60 min at 60°C to a solution of 2.43 g (0.01M) 4-nitro-naphthalic anhydride in ethanol, then stirred for another 60 min under these conditions. After cooling, the liquid has been added to 200ml water and the precipitate has been filtered out. Yield 79%, 2.61g.
3.3. Synthesis of 6-(dimethylamino)-2-(pyridin-4-ylmethyl)-1H-benzo[de]isoquinoline-1,3(2H)-dione (NI2)
The compound NI 0 (1.65 g ,0.005 M) was dissolved in 20 ml of DMF, and 0.101 ml (0.005 M) of dimethylamine was added and after that and the mixture is stirred for 24 250 hours at 25 ° C. The liquid was poured into 200ml water and filtered the formed precipitate. Yield 81%, 1.33g.”
Answer:
The IR, 1H-NMR and 13C-NMR spectra of the synthesized compounds NO, N1 and N2 are added in Supporting information.
2. Part of the discussion should be completed.
3. Whether the authors were tempted to do statistical analysis?
Answer:
Statistical analysis was performed only when the microbiological activity of the compounds and the cotton fabrics dyed with them was examined.
Reviewer 3 Report
1,8-naphthalimides are novel bioactive compounds with good antibacterial activity and could be used as photosensitizers in photodynamic therapy. In the current study, the authors synthesized two reactive 1,8-naphthalimide derivatives (NI1 and NI2) and characterized their photophysical characteristics, bactericidal activity. In addition, the 1,8-naphthalimide derivatives were also covalently bonded to the surface of cotton fabrics. The novel findings showed that both compounds have good antibacterial activity against bacteria. I think the results are reliable, which can provide new ideas and evidence for the development and utilization of novel photosensitizers and antibacterial agents.
Here are several questions and/or suggestions the authors may consider.
1. My major concern is the authors should provide more introduction about the parent molecule 1,8-naphthalimides. Since I did not find the comparation between 1,8-naphthalimides and NI1/ NI2 about their photodynamic activity and bactericidal activity, therefore, the authors should provide more evidence to justify their study.
2. The Figure 1 seems to be a little blur, please kindly increase its clarity;
3. In figure 3, it seems the label of the subfigure (B) is missing.
4. The English should be proofread before resubmission. For example, Line 179 and Line 183 “in the presenece ot NI2” Should be “of”, not “ot”. Line 181 “(2) (B).” Should be (B).
5. The editing mistakes should be avoided in the article: The title formats of the figures were not uniform; there is no space before paragraph in Line 278, 284 and 293.
Author Response
1,8-naphthalimides are novel bioactive compounds with good antibacterial activity and could be used as photosensitizers in photodynamic therapy. In the current study, the authors synthesized two reactive 1,8-naphthalimide derivatives (NI1 and NI2) and characterized their photophysical characteristics, bactericidal activity. In addition, the 1,8-naphthalimide derivatives were also covalently bonded to the surface of cotton fabrics. The novel findings showed that both compounds have good antibacterial activity against bacteria. I think the results are reliable, which can provide new ideas and evidence for the development and utilization of novel photosensitizers and antibacterial agents.
Here are several questions and/or suggestions the authors may consider
My major concern is the authors should provide more introduction about the parent molecule 1,8-naphthalimides. Since I did not find the comparation between 1,8-naphthalimides and NI1/ NI2 about their photodynamic activity and bactericidal activity, therefore, the authors should provide more evidence to justify their study.
The study of 1,8-naphthalimides as PS has been described by us recently where they have been attached to the dendrimers periphery. In this work, for the first time, we investigate covalently bound 1,8-naphthalimide to cotton fabric and its photodynamic antibacterial activity. Some reviews are cited that describe the use of 1,8-naphthalimides with microbiological activity. Direct comparison of such activity is difficult for the reason that the experiments are conducted under different conditions.
For this reason, we also do not give such a comparison.
2. The Figure 1 seems to be a little blur, please kindly increase its clarity;
Figure 1 was corrected
- In figure 3, it seems the label of the subfigure (B) is missing.
The label for Fig 3 was corrected.
- The English should be proofread before resubmission. For example, Line 179 and Line 183 “in the presenece ot NI2” Should be “of”, not “ot”. Line 181 “(2) (B).” Should be (B).
It was corrected
- The editing mistakes should be avoided in the article:The title formats of the figures were not uniform; there is no space before paragraph in Line 278, 284 and 293.
It was corrected
Round 2
Reviewer 1 Report
In the response to the prior review, the authors often claim that the text contains answers, but these are often nowhere to be found. Laundering followed perhaps by the autoclave is commonly used to decontaminate fabrics in a hospital setting. What unmet need is this project designed to solve? There are microorganisms resistant to the autoclave? What might be the volume of fabrics that would need decontamination in a hospital setting, what size vats of boiling ethanol will be needed and how will these fabrics be irradiated (on both sides)? A solar simulator is clearly unequal to the task.
There is still no information on exactly how bacteria are applied to these fabrics, how they are irradiated and how tests for efficacy are carried out. Under exactly what conditions was a bacterial culture applied? How did irradiation occur? This entire section is vague and lacking detail.
This has every aspect of a cure for which there is no disease. As pointed out before, fluorescence of the dyes is not relevant to the argument. Legends to Figs. 5 and 6 contain no information on the nature of irradiation (wavelength, light dose), Other experimental details are lacking. The procedures described (to the extent that they are described) are adequate only for 1 gram sample sizes. Most hospitals use larger fabrics.
Author Response
In the response to the prior review, the authors often claim that the text contains answers, but these are often nowhere to be found. Laundering followed perhaps by the autoclave is commonly used to decontaminate fabrics in a hospital setting. What unmet need is this project designed to solve? There are microorganisms resistant to the autoclave? What might be the volume of fabrics that would need decontamination in a hospital setting, what size vats of boiling ethanol will be needed and how will these fabrics be irradiated (on both sides)? A solar simulator is clearly unequal to the task.
As the reviewer mentioned, one possible application of this new textile material is in a hospital environment. However, frequent washing of all hospital textiles is not always easy and sometimes even impossible. Antimicrobial textiles are therefore a well-known solution to this problem. In recent years, especially after the Covid pandemic, new technology has become necessary to reduce microbial contamination based on UV light disinfection systems. However, this requires special equipment and appropriate precautions to protect people.
The use of visible light or light from an ordinary artificial light source to sterilize textiles has the potential to solve the problem, which is the aim of our research.
There is still no information on exactly how bacteria are applied to these fabrics, how they are irradiated and how tests for efficacy are carried out. Under exactly what conditions was a bacterial culture applied? How did irradiation occur? This entire section is vague and lacking detail.
The antimicrobial activity of cotton fabrics treated with the compounds NI1 and NI2 was tested in MPB containing Gram-positive Bacillus cereus or Gram-negative Pseudomonas aeruginosa under light irradiation and in dark. Tubes containing MPB and square shape cotton speciments (10 mm x 10 mm) were inoculated with standardized microbial suspension. Tubes with native cotton and without speciments were also prepared as controls. Two sets of tubes were prepared for the tests in the presence and absence of light. After 18 h incubation at an appropriate temperature, the specimens were removed and OD600 was determined. The antimicrobial activity of the samples was evaluated by the reduction of bacterial growth in presence of the treated speciments compared to native. All assays were performed in triplicate and the averages are given (standard deviations less than 5%).
A Xe lamp 150 W, with a spectral window of the solar simulator: 185-1100 nm has been used for irradiation of all antibacterial studies at a distance of 50 cm to the samples.
This has every aspect of a cure for which there is no disease. As pointed out before, fluorescence of the dyes is not relevant to the argument. Legends to Figs. 5 and 6 contain no information on the nature of irradiation (wavelength, light dose), Other experimental details are lacking. The procedures described (to the extent that they are described) are adequate only for 1 gram sample sizes. Most hospitals use larger fabrics.
Regarding antibacterial photodynamic therapy, fluorescence is important because in this case energy transfer of the molecules from excited to the ground state is important. Irradiation conditions are given in the experimental part, therefore, in order not to confuse the legends of the figures, they are not given. We have used 1cm2 fabrics to demonstrate the activity of our molecules and textile materials.
Reviewer 2 Report
Should be corrected:
Figure S2. 13H-NMR spectrum of NI0.
Figure S4. 13H-NMR spectrum of NI1.
Figure S6. 13H-NMR spectrum of NI2.
Author Response
Figure S2. 13H-NMR spectrum of NI0.
Figure S4. 13H-NMR spectrum of NI1.
Figure S6. 13H-NMR spectrum of NI2.
The legends of Figures S2, S4, and S6 were corrected as 13C-NMR spectrum of NIs